# Camouflage Correction of Skeletal Class III Severe Open Bite with Tooth Ankylosis Treated by Temporary Anchorage Devices: A Case Report

**DOI:** 10.3390/dj11040107

**Published:** 2023-04-21

**Authors:** Yuka Yashima, Masato Kaku, Taeko Yamamoto, Cynthia Concepcion Medina, Shigehiro Ono, Yosuke Takeda, Kotaro Tanimoto

**Affiliations:** 1Department of Orthodontics and Craniofacial Developmental Biology, Graduate School of Biomedical Sciences, Hiroshima University, 1-2-3 Kasumi, Minami-ku, Hiroshima 734-0037, Japan; 2Department of Anatomy and Functional Restorations, Division of Oral Health Sciences, Graduate School of Biomedical Sciences, Hiroshima University, 1-2-3 Kasumi, Minami-ku, Hiroshima 734-0037, Japan; 3Department of Oral and Maxillofacial Surgery, Graduate School of Biomedical Sciences, Hiroshima University, 1-2-3 Kasumi, Minami-ku, Hiroshima 734-0037, Japan; 4Dental Practitioner, Yumemirai Dental & Orthodontic Clinic Saijo, 1172 Sukezane, Saijo, Higashi Hiroshima 739-0021, Japan

**Keywords:** tooth ankylosis, open bite, surgical luxation, TAD, skeletal class III, camouflage treatment

## Abstract

Tooth ankylosis is a disorder characterized by the fusion of tooth and alveolar bone. This case report describes the treatment of a severe open bite due to tooth ankylosis. A 14-year-old female patient with a chief complaint of masticatory dysfunction was diagnosed with skeletal Class III severe anterior open bite and tooth ankylosis. She visited our university hospital with a chief complaint of an anterior open bite. After the surgical luxation of the ankylosed maxillary right central incisor, the tooth was orthodontically retracted using a nickel-titanium wire. The right mandibular lateral incisor and canine were luxated and retracted using intermaxillary elastics from a temporary anchorage device (TAD), which was inserted in the opposite jaw. During the treatment, skeletal Class III malocclusion deteriorated due to anterior growth of the mandible. Therefore, TADs were inserted into the retromolar pad on both sides of the mandible and retracted into the mandibular dental arch. Although the mandibular right canine was luxated several times, it could not be brought to the occlusal line, and was thus extracted; the extraction space was replaced with a prosthesis. Consequently, a normal overjet and overbite with a straight profile were achieved. Extrusion of ankylosed teeth by intermaxillary elastics from a TAD is a valid treatment option for patients with severe open bites.

## 1. Introduction

Tooth ankylosis is generally characterized by the fusion of bony structures with one another [1]. This disorder is defined as the obliteration of the periodontal ligament (PDL); thus, the involved tooth is fused to the surrounding bone, preventing eruption and orthodontic movement [1]. The dental ankylosis of permanent teeth is 10 times less frequent than primary teeth, with mandibular and maxillary first molars being most frequently ankylosed followed by maxillary canines and incisors [2]. In orthodontics, misdiagnosis or lack of recognition of ankylosed teeth leads to compromised treatment progress and results. Unfortunately, ankylosed teeth are difficult to diagnose, particularly in the early stages. Key features for diagnosing dental ankylosis include a lack of mobility and orthodontic movement, a dull sound on percussion, tipping of adjacent teeth, ditching of the alveolar process, and severely infra-positioned occlusion. The frequency of ankylosis occurrence is 22.4% in Sweden [3], and permanent tooth ankylosis is reported to occur at approximately 10% of the frequency with which it occurs in deciduous teeth [4]. Ankylosis can be described as punctate, planar, or three-dimensional, in order of increasing severity; if progression occurs from one form to another, it may proceed slowly or rapidly. Ankylosis is caused by several factors including local metabolic changes, genetic predisposition, dental trauma, dislocation disorder, apical infection, post-excision transplantation, scientific or thermal stimulation, and surgical orthodontic treatment [5]. Clinical signs include a cusp tip/incisal edge lower than those of the adjacent teeth, indistinguishability between the root and surrounding bone in the PDL as observed on radiographs, and clear percussion sound [6]. However, the percussion sound is difficult to objectively evaluate because it is affected by the mass of the tooth and the physical normality of the periodontal tissue. It is also difficult to recognize bone adhesions on radiographs [7], and only approximately 30% of ankylosed teeth can be identified radiographically from animal experiments [8]. 

Recently, newer diagnostic tools have been introduced, namely digital soundwave and resonance frequency analyses [9,10]. However, because of fluctuating sensitivity and poor acceptance among patients, these tools have seen limited use in clinical practice. Therefore, percussion and mobility testing are still considered the primary diagnostic tools, and many researchers agree that it is almost impossible to determine tooth ankylosis using any method other than confirming the degree of dental movement [8,11,12].

Orthodontic treatments for tooth ankylosis include: (1) retraction after luxation [13]; (2) alveolar osteotomy [14,15,16,17,18]; (3) alveolar osteotomy and bone distraction [19,20,21,22,23,24]; (4) alveolar osteotomy and autologous transplantation [25]; and (5) alveolar osteotomy and repositioning methods [26]. Among these, retraction after luxation is frequently used because it is surgically less invasive [13]. However, even when tooth retraction is performed after luxation, the affected tooth may reattach after some time and unexpected reactions may occur [13]. In this case report, we aim to describe a case of ankylosed tooth resolved by using temporary anchorage devices (TADs) after luxation, with no adverse reactions.

## 2. Materials and Methods

### 2.1. Chief Complaints

A Japanese girl aged 14 years and 7 months visited the Hiroshima University Hospital with a chief complaint, “I cannot bite anything with my front teeth”.

### 2.2. History of Present Illness

Her pretreatment facial appearance revealed a concave profile and the mandible shifted to the right side (Figure 1). The initial intraoral photographs (EOS R, CANON, Tokyo, Japan) revealed a 0.5 mm overjet and a −6.0 mm anterior open bite with occlusal contact only between the first and second molars on the right side. The molar occlusal relationship was classified as Class I. There was no crowding in the upper arch, but some spaces were found in the lower arch. The lower dental midline had shifted to the right by 4.0 mm, whereas the upper midline coincided with the facial midline (Figure 2). 

### 2.3. History of Past Illness

She fell on the ground and landed on her upper and lower right anterior teeth when she was 9 years old. Particularly, she severely hit the upper right central incisor and lower right canine region. The patient did not have any history of systemic illness. 

### 2.4. Personal and Family History

She and her parents did not have any history of systemic illness.

### 2.5. Clinical Examination

The patient’s oral hygiene was poor (plaque control record: 67%, probing pocket depth was less than 3 mm without bleeding on probing). Therefore, continuous teeth brushing instruction was performed. To test the mobility of teeth around the open bite region, a resonance frequency analyzer was used (Periotest, Gulden-Medizinteknik, Eschenweg, Modautal, Germany). Periotestometry was repeated 3 times for each tooth and the mean values were shown in Table 1. The Periotest values (PTV) were found to be small in the upper right central incisor and the lower right canine and lateral incisor. Therefore, the upper right central incisor and the lower right canine and lateral incisor were identified as ankylosed. The PTV of the lower right incisor was smaller than that of the lower left incisor (Table 1). 

### 2.6. Imaging Examinations

A panoramic radiograph (Hyper-X, Asahi Roentgen, Kyoto, Japan) revealed loss of the PDL cavity in the upper right central incisor, lower right lateral incisor, and lower right canine. Although the root of the upper and lower anterior tooth was relatively short, the association with the ankylosis was not clear. There was no abnormal condition in both the right and left temporomandibular joint (Figure 3). A skeletal Class III relationship of the angle of point A-nasion-point B (ANB; 0.2°) and a steep Frankfort-mandibular plane angle (FMA; 36.2°) with lingual inclination of the lower incisors of incisor-mandibular plane angle (IMPA; 79.5°) were noted in the Hiroshima University cephalometric analyses (CX-150 W, Asahi Roentgen, Kyoto, Japan, Figure 4 and Table 2).

### 2.7. Diagnosis

Based on this information, this patient was diagnosed to have a skeletal Class III open bite with mandibular deviation, with ankylosis of the upper right central incisor and lower right canine and lateral incisor, in addition to a spaced lower arch. She was recommended orthodontic treatment with orthognathic surgery, but the patient did not accept receiving the surgery.

Informed consent was obtained from the patient and her parents prior to the start of orthodontic treatment.

The treatment objectives were as follows: (1) to diagnose which tooth is actually ankylosed; (2) to retract the ankylosed teeth via luxation; (3) to achieve normal overjet and overbite with ideal occlusion. The treatment plan was as follows: Fixed rigid lingual arch devices were cemented on both the upper and lower jaws to assist with both anchorage and eruption of the ankylosed teeth.All teeth were moved orthodontically to diagnose the ankylosed teeth.Surgical luxation was performed on the ankylosed maxillary right central incisor, followed by orthodontic retraction using a nickel-titanium (NiTi) wire.The right mandibular lateral incisor and canine were luxated and retracted by intermaxillary elastics from a TAD that was inserted on the buccal alveolar bone between the right maxillary lateral incisor and canine.

### 2.8. Treatment Procedures

After bonding the brackets to all teeth (Clear Bracket, Metal Bracket, and Buccal Tube, JM Ortho, Tokyo, Japan), leveling was initiated to alleviate the lower occlusal plane on both the upper and lower dentitions. Lingual arches were set on both the upper and lower dental arches to assist with the anchorage and to prevent further distortion of the occlusal planes.

After initial leveling of the maxillary dentition with a 0.016-inch NiTi (Sentalloy, Tomy International, Tokyo, Japan), the upper right central incisor, lower right lateral incisor, and lower right canine were diagnosed as ankylosed teeth because they did not show tooth movement. Then, the tooth was luxated and a 0.014-inch NiTi (Sentalloy, Tomy International, Tokyo, Japan) wire was used to retract the ankylosed teeth (Figure 5). Next, the wire was changed to a 0.016 × 0.022-inch Co-Cr wire (Blue Elgiloy JM Ortho, Tokyo, Japan). The lower right canine was retracted toward the occlusal plane using the lower lingual arch with an attached hook as an anchorage point. Six months after beginning the treatment, the overjet changed to −2.0 mm with both Angle Class III molar relationships due to the flattening of the lower occlusal plane and the anterior growth of the mandible. To achieve appropriate overjet and overbite, TADs (1.6 mm in diameter and 8 mm in length, Dual Top Auto Screw; Jeil Medical Corp., Seoul, Republic of Korea) were implanted bilaterally on the mandibular retromolar pads to retract the lower dental arch. The TAD was also inserted on the buccal side of the alveolar bone adjacent to the upper right lateral incisor and canine to retract the ankylosed teeth using intermaxillary elastics (Figure 6 and Figure 7). Surgical luxation of the ankylosed teeth was performed using anterior after local anesthesia. Luxation of the teeth was repeated several times whenever the teeth showed ankylosis. However, because the lower right canine erupted only halfway to the occlusal plane, the tooth was extracted and after healing the periodontal tissue, the resulting space was closed with a ceramic crown using the right lateral incisor as a single retainer. After 50 months of active orthodontic treatment, all orthodontic appliances, including TADs, were removed, and bonded lingual retainers were adapted on the lingual side of both the upper and lower anterior dentition (from upper right canine to left canine, and from lower right lateral incisor to left canine, Penta Cat Wire, Tomy International, Tokyo, Japan). Moreover, a wraparound retainer was placed on the upper and lower dentitions.

## 3. Results

After orthodontic treatment, the arch alignment was well corrected, the overbite increased to 2.2 mm, and the overjet improved to 2.0 mm. There was no major change in the post-treatment facial profile compared to the pretreatment profile. The left molar relationship was Angle Class I, whereas that on the right was Angle Class III. The space of lower right canine was repaired by direct bonding right after the debonding of all orthodontic appliances. The mandibular dental midline right shift changed to 2.0 mm (Figure 8 and Figure 9). The post-treatment panoramic radiograph did not show severe root resorption (Figure 10). On the cephalometric pre-treatment and post-treatment superimposition, the ANB angle had changed from 0.2° to −1.3°. The mandibular anterior teeth tipped lingually (FMIA; from 64.3° to 74.6°) in order to obtain an appropriate overjet. U1 to A-Pog decreased from 7.7 mm to 6.2 mm, and L1 to B-Pog decreased from 7.1 mm to 4.0 mm. Before and after the treatment, the upper lip changed from 0.1 mm to −0.9 mm and the lower lip changed from 2.4 mm to 0.1 mm compared to the E-line (Figure 11, Figure 12 and Figure 13, Table 2).

## 4. Discussion

Ankylosis is a direct connection between a tooth and the alveolar bone without the intervening PDL, and ankylosed teeth cannot show natural eruption and orthodontic tooth movement. The criteria for determining how to deal with ankylosed teeth are as follows: the patient’s growth stage, indication for a tooth extraction case or a non-extraction case as an orthodontic treatment, and the types of teeth involved and the degree of tooth ankylosis. As orthodontic treatments, some approaches to correct ankylosed teeth include retraction after luxation [13], alveolar osteotomy [14,15,16,17,18], tooth extraction, and preservation (conservative treatment or replacement of the extraction space with a prosthesis). Because the patient, in this case, was in a growing stage, an alveolar osteotomy could not be performed. Moreover, with sufficient space to align all teeth, luxation of ankylosed teeth and retraction were chosen. Tooth retraction after luxation is frequently used for ankylosed teeth because it is less surgically invasive than other treatments, and many successful cases have been reported [27,28,29,30,31]. In this case, three ankylosed teeth were identified, all of which were moved by retraction after luxation. Although intermaxillary elastics have often been used for an open bite, the extent of the open bite, in this case, was too large. Furthermore, there was a possibility that an adverse reaction to the opposite jaw could occur, such as distortion of the occlusal plane due to ankylosed teeth. Therefore, the TAD was set on the opposite jaw to use intermaxillary elastics. Previous reports showed that the areas between the first and second premolars in the maxilla had the highest success rate of TADs [32]. These results suggested that root proximity might affect the success rate of TADs. In the present case, TAD was inserted on the buccal side of the alveolar bone adjacent to the upper right lateral incisor and canine which has sufficient space for the insertion of TAD. Sfondrini et al. demonstrated that a minimum TAD diameter of 1.7 mm could be considered in order to reduce risks of bending and fracture. However, it was also shown that there were no significant differences between diameters of 1.6 mm and 1.7 mm, for both bending and maximum loads [33]. Therefore, in the present case, we used TADs with 1.6 mm in diameter. Resultantly, we succeeded in preventing adverse reactions during the retraction of the ankylosed teeth. However, although the right lower canine was luxated several times, the tooth erupted to only half of the adjacent tooth length and could not reach the occlusal line; thus, the tooth was extracted, and prosthetic treatment was administered. 

To anchor the prosthesis, we applied direct bonding prosthesis because the patient’s satisfaction was greater than those with a removable partial denture in young people. As it is expected that vertical development of natural teeth in young females will occur even after adulthood, a dental implant could not be applied at this time [34]. Pjetursson showed that survival rates of both dental implants and restorations in a combined tooth–implant-supported prosthesis were lower than those in a solely implant-supported prosthesis [35]. Hence, they recommended that prosthetic rehabilitation should be applied with a solely implant-supported prosthesis. In this case, since the lower right lateral incisor was the ankylosed tooth, we decided to set direct bonding prosthesis as a single retainer because the difference in the degree of displacement between the healthy right first premolar and the ankylosed lateral incisor is the same as dental implants. The prosthetic device was esthetically pleasing, and patient satisfaction was very high, although this device did not participate in the occlusive relationships in order to avoid debonding and fracturing. Therefore, we are going to replace this device with a dental implant to improve occlusal function after the patient’s skeletal changes are complete.

In this case, the treatment period was longer than four years because residual growth of the mandible occurred, and retraction of the mandibular molars was required during treatment. Orthodontic treatment for most skeletal Class III patients should be initiated after the end of the major period of mandibular growth. However, early treatment was required to retract the ankylosed teeth, and long-term treatment was required in this case. To achieve appropriate overjet and overbite, TADs were implanted bilaterally on the mandibular retromolar pads to retract the lower dental arch. Previous systematic reviews demonstrated that the most common complication with the insertion of TAD is the occurrence of lesions at the root during interradicular insertion and perforation of the maxillary sinus and nasal cavity [36]. As mandibular retromolar pads are the safety area to avoid root injuries and they are the appropriate site to allow correct force direction for retraction of the lower dental arch, final camouflage treatment could be achieved in the present case. Sometimes, skeletal Class III patients cannot obtain facial esthetics and ideal occlusion with camouflage orthodontic treatment. In the present case, the best treatment plan would be the combination of orthodontics and orthognathic surgery to achieve an ideal occlusion and good esthetic result. However, the treatment result of the present case showed remarkable improvement in occlusal function, with total patient satisfaction. Therefore, it is concluded that the use of TADs was useful for the retraction of ankylosed teeth and distalization of the lower arch for camouflage skeletal Class III treatment without any adverse reactions. However, a long-term follow-up will be necessary for occlusal stability and root resorption of ankylosed teeth by radiograph examination.

## Figures and Tables

**Figure 1 dentistry-11-00107-f001:**
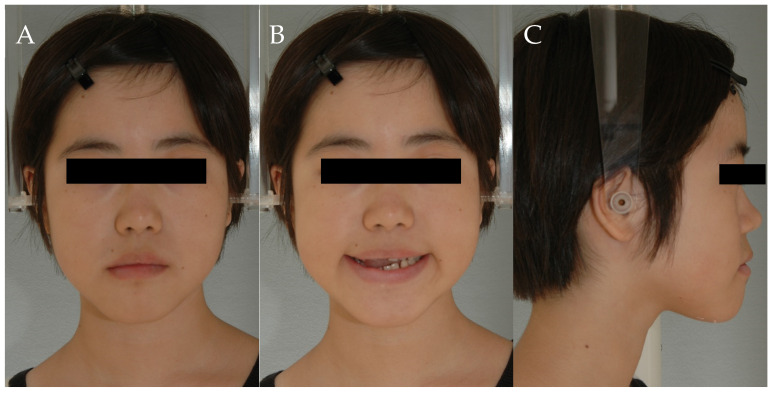
Pre-treatment facial photographs. (**A**) Front view; (**B**) smile view; (**C**) lateral view.

**Figure 2 dentistry-11-00107-f002:**
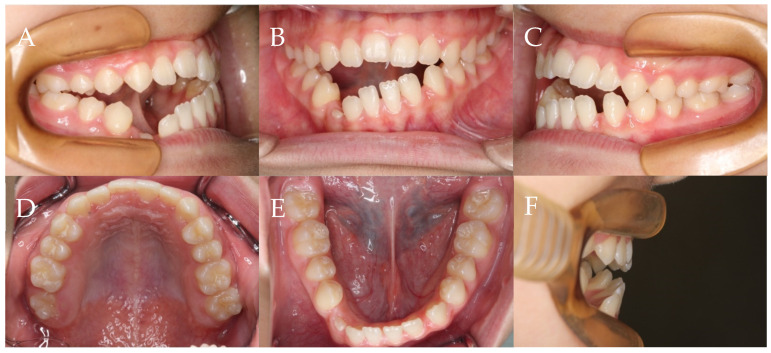
Pre-treatment intraoral photographs. (**A**) Right side view; (**B**) front view; (**C**) left side view; (**D**) upper occlusal view; (**E**) lower occlusal view; (**F**) incisal view.

**Figure 3 dentistry-11-00107-f003:**
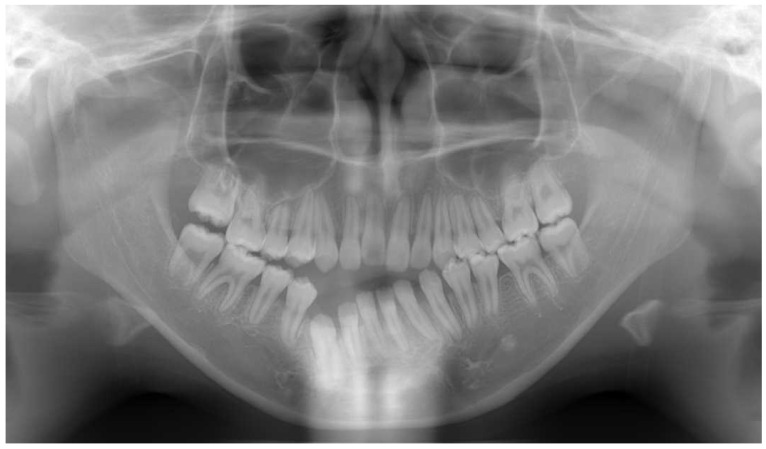
Pre-treatment panoramic radiograph.

**Figure 4 dentistry-11-00107-f004:**
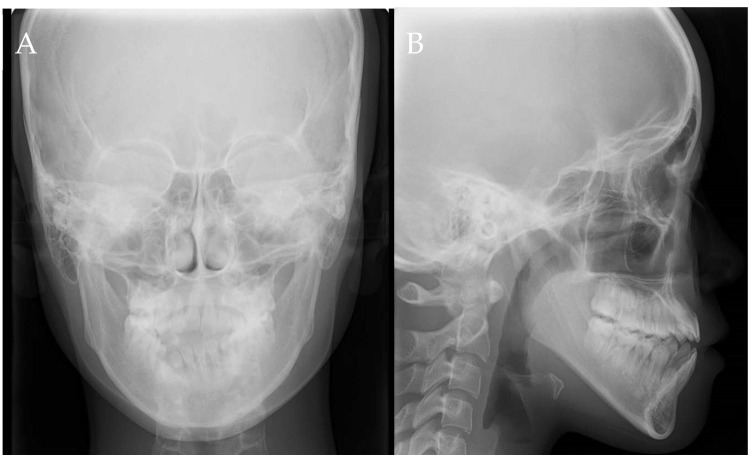
Pre-treatment cephalometric radiograph. (**A**) Front view; (**B**) lateral view.

**Figure 5 dentistry-11-00107-f005:**
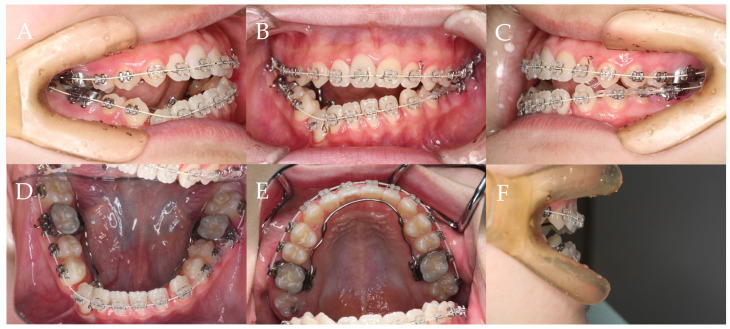
Leveling after surgical luxation of ankylosed teeth. (**A**) Right side view; (**B**) front view; (**C**) left side view; (**D**) upper occlusal view; (**E**) lower occlusal view; (**F**) incisal view.

**Figure 6 dentistry-11-00107-f006:**
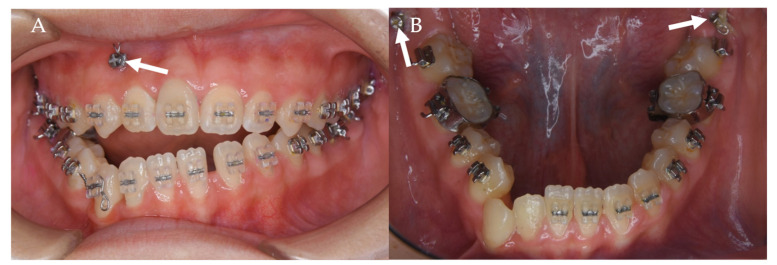
Intraoral photographs after insertion of TADs. (**A**) front view; (**B**) lower occlusal view. Arrows indicate TADs.

**Figure 7 dentistry-11-00107-f007:**
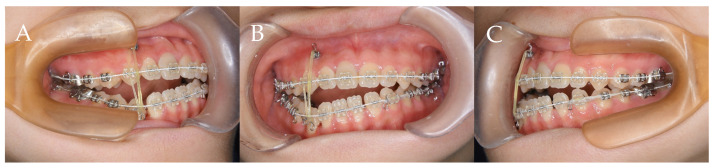
Retraction of the ankylosed lower right canine using intermaxillary elastics. (**A**) Right side view; (**B**) front view; (**C**) left side view.

**Figure 8 dentistry-11-00107-f008:**
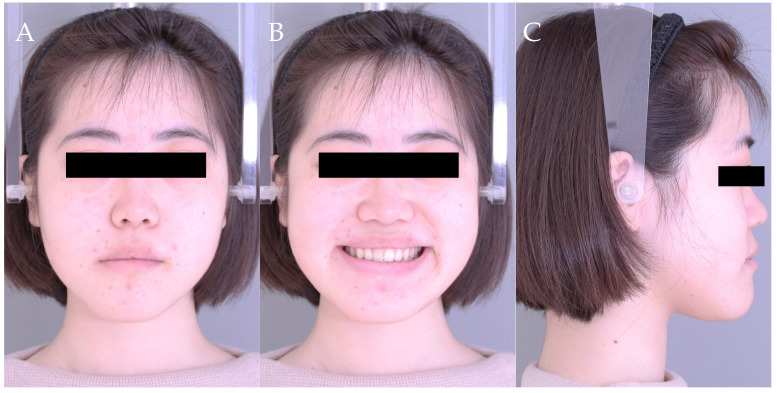
Post-treatment facial photographs. (**A**) Front view; (**B**) smile view; (**C**) lateral view.

**Figure 9 dentistry-11-00107-f009:**
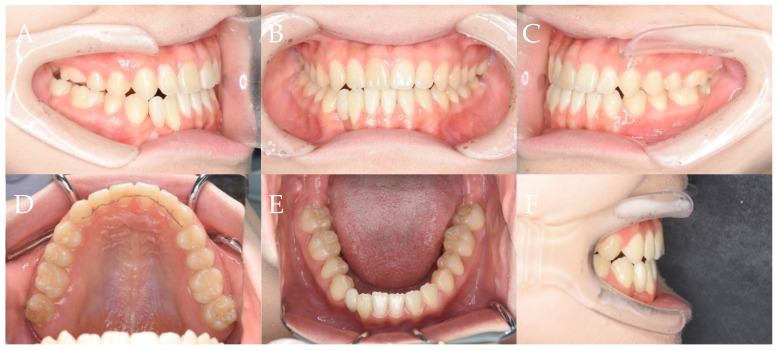
Post-treatment intraoral photographs right after debonding all orthodontic appliances. (**A**) Right side view; (**B**) front view; (**C**) left side view; (**D**) upper occlusal view; (**E**) lower occlusal view; (**F**) incisal view.

**Figure 10 dentistry-11-00107-f010:**
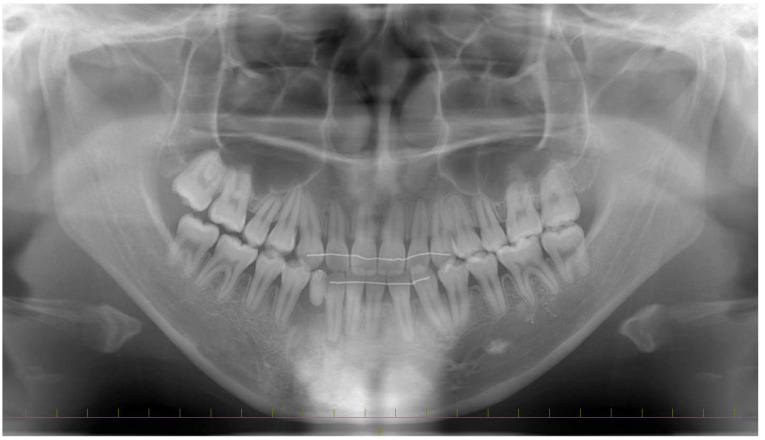
Post-treatment panoramic radiograph.

**Figure 11 dentistry-11-00107-f011:**
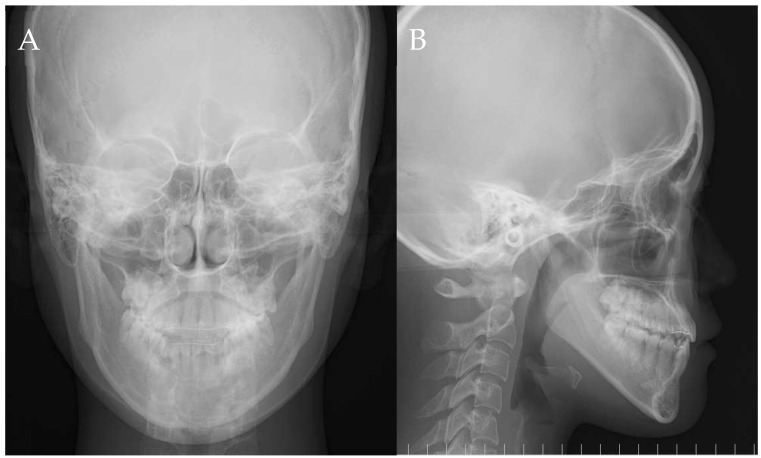
Post-treatment cephalometric radiograph. (**A**) Front view; (**B**) lateral view.

**Figure 12 dentistry-11-00107-f012:**
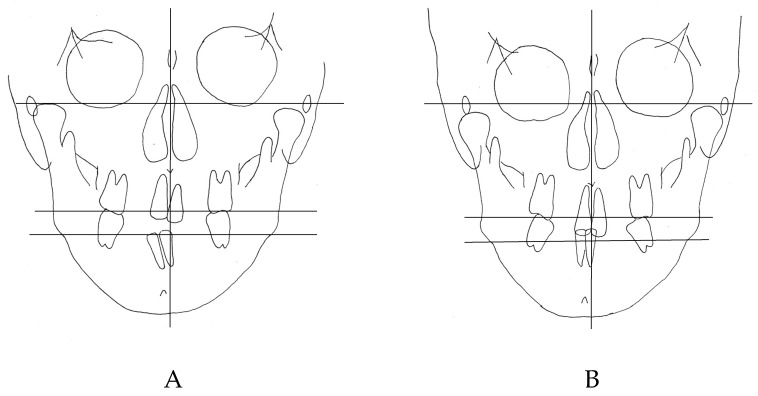
Posterior–anterior cephalometric tracing. (**A**) Pre-treatment (14 y 7 m); (**B**) post-treatment (18 y 1 m).

**Figure 13 dentistry-11-00107-f013:**
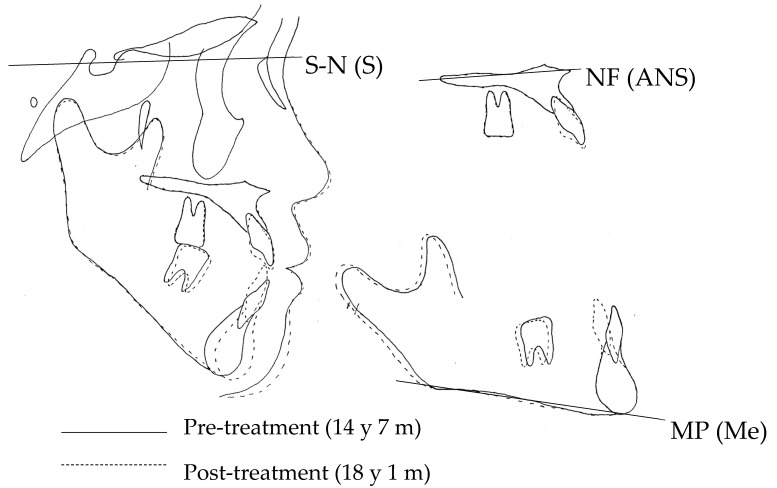
Lateral cephalometric superimposition. S-N (S): Superimposition on sella-nasion at sella; MP (Me): superimposition on the mandibular plane at menton; NF (ANS): superimposition on the nasal floor at anterior nasal spine.

**Table 1 dentistry-11-00107-t001:** Periotest value.

Tooth (FDI System)	13	12	11	31	32	41	42	43	44
Periotest value (PTV)	8.0	17.5	1.7	19.9	20.0	17.6	4.5	−1.7	5.1

**Table 2 dentistry-11-00107-t002:** Summary of cephalometric measurements.

Measurements	Pre-Treatment	Post-Treatment
SNA (°)	81.5	81.5
SNB (°)	81.3	82.8
ANB (°)	0.2	−1.3
FMA (°)	36.2	36.5
FMIA (°)	64.3	74.6
IMPA (°)	79.5	68.9
U1-SN (°)	110.9	106.9
U1 to A-Pog (mm)	7.7	6.2
L1 to B-Pog (mm)	7.1	4.0
E-line: Upper (mm)	0.1	−0.9
E-line: Lower (mm)	2.4	0.1

## Data Availability

Not applicable.

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
