# Peer review of "Camouflage Correction of Skeletal Class III Severe Open Bite with Tooth Ankylosis Treated by Temporary Anchorage Devices: A Case Report"

_dentistry, 2023, doi:10.3390/dj11040107_

Round 1

Reviewer 1 Report

Dear authors!

Thank you for presenting a very interesting clinical case. The study is well structured and the content is clearly stated. However, some questions appeared after getting acquainted with the manuscript. On these issues more detailed information should be given.

1.     History of past illness.

It is believed that trauma is one of the main reasons of tooth ankylosis. In the history of patient’s past illness, you indicated a trauma of tooth 11.

1.1.Please comment on tooth 33 which was identified as ankylosed during diagnostic examinations.

1.2. It is advisable noting approximate age of patient at the moment of tooth trauma.

2.     Oral hygiene.

Please specify which method was used to measure the patient’s hygiene level.

3.     Periotestometry.

3.1. Please specify the methodology of periotestometry. Despite the advantages of this method there are disadvantages related to data accuracy. To minimize errors, it is desirable to perform periotestometry several times (at least 3 times).

3.2. Please clarify why in the table 1 there is no information about remaining teeth. From my point of view, this information will not be superfluous.

4.     Diagnostics.

4.1. Diagnosis of ankylosed teeth is a difficult task. The mentioned and used methods such as percussion, periotestometry and X-ray diagnostics are not reliable enough, since they are associated with the probability of establishing false positive or false negative conclusions. The most reliable method is considered to be the application of orthodontic force, but you have not provided information about this method. Please give a comment on this.

5.     Treatment.

The method used for replacing the missing tooth 33 is very interesting. Insufficient attention is paid to this aspect of treatment in the study. I could not find answers to the following questions in the text of the article:

5.1. Why such method of missing tooth replacement was used in the study? For example, the removable construction could be used as alternative method. Removable construction also would allow to keep the space between adjacent teeth for the further implantation.

5.2. Are you going to replace the missing tooth with artificial one supported by dental implant in the future? Please complete the text with information about the long-term treatment plan.

5.3. Why was the tooth 32 chosen as a supporting not tooth 34? From the point of view of masticatory force redistribution, tooth 34 or two adjacent teeth (34 and 32) are worth using as supporting structures.

5.4. In the text [lines 289-290] you indicated that tooth 34 was healthy so a tooth 32 was used as supporting. However, a tooth 32 was also healthy. Such logic is not clear enough for me. Please give additional arguments.

5.5. It is unclear how the reliability of the method of replacement of the missing tooth 33 is achieved. You have mentioned that the tooth is made of ceramic material that was simply attached to the lateral surface of the tooth 32. In my opinion, such adhesive bonding is not reliable. Have there been cases of artificial tooth debonding or fracturing? Which fixation protocol was used? Does tooth 33 participate in occlusive relationships? Please give comments on this.

5.6. In the text [line 291] you conclude that the artificial crown 33 allowed to restore the function, but the article does not provide any objective data confirming this fact. In my opinion, in order to reduce the risk of technical complications of such treatment, it would be necessary to remove the artificial tooth 33 from the occlusive relationship or significantly reduce the masticatory load that falls on it, which cannot be considered the restoration of masticatory function. The wording needs to be corrected.

5.7. Please comment on the post-treatment panoramic radiograph. How do you assess the short-term and long-term results of the treatment?

6.     Conclusions.

Unfortunately, one clinical case does not provide grounds for formulating a conclusion about the effectiveness of treatment. The format of the clinical case is descriptive, in which the facts are simply stated. For example, in CARE (https://www.care-statement.org/checklist) there is no requirement to formulate conclusions, and therefore I would recommend deleting the section "Conclusions" from the article.

Author Response

April 10, 2023

Dear Editor, Dentistry Journal

I have now completed to revise and am now sending the manuscript entitled “Correction of skeletal Class III severe open bite with tooth ankyloses treated by temporary anchorage devices: A case report” (No: 2323350). I have revised the manuscript in accordance with the referee’s suggestions. A list of annotations is shown below. I hope the revised version will be well accepted by the editorial committee.

Thank you very much for all the troubles you've taken for me.

Sincerely Yours,

Yuka Yashima, Masato Kaku, Taeko Yamamoto, Cynthia Concepcion Medina, Shigehiro Ono, Yosuke Takeda and Kotaro Tanimoto

Address correspondence to: Masato Kaku, D.D.S., PhD.

Department of Anatomy and Functional Restorations,

Division of Oral Health Sciences,
Hiroshima University Graduate School of Biomedical and Health Sciences, Kasumi, Minami-ku, Hiroshima 734-8553, Japan.

E-mail: mkaku@hiroshima-u.ac.jp

To Reviewer 1

  1. History of past illness.

It is believed that trauma is one of the main reasons of tooth ankylosis. In the history of patient’s past illness, you indicated a trauma of tooth 11.

  • Please comment on tooth 33 which was identified as ankylosed during diagnostic examinations.

Thank you for your suggestion. We added the sentence for lower right canine as ankylosed in physical examination.

Therefore, the upper right central incisor and the lower right canine and lateral incisor were identified as ankylosed.

1.2. It is advisable noting approximate age of patient at the moment of tooth trauma.

Thank you very much for your suggestion.

We added the sentence as below in the History of past illness

She had bruised her upper and lower right anterior teeth when she was 9 years old. Particularly, she strongly hit the upper right central incisor and lower right canine region.

  1. Oral hygiene.

Please specify which method was used to measure the patient’s hygiene level.

Thank you for your suggestion. We are very sorry that the patient’s oral hygiene was not good actually. So, we added the sentence in the Physical examination (2.5.) as below.

The patient’s oral hygiene was poor (plaque control record: 67%, probing pocket depth are less than 3 mm without bleeding on probing). So, continuous teeth brushing instruction was performed.

  1. Periotestometry.

3.1. Please specify the methodology of periotestometry. Despite the advantages of this method there are disadvantages related to data accuracy. To minimize errors, it is desirable to perform periotestometry several times (at least 3 times).

3.2. Please clarify why in the table 1 there is no information about remaining teeth. From my point of view, this information will not be superfluous.

Thank you for your suggestion. Unfortunately, we applied periotestometry only on the teeth of open bite region. So, we added the sentence as below in the clinical examination.

2.5. Clinical examination

To test the mobility of teeth around open bite region, a resonance frequency analyzer was used (Periotest, Gulden-Medizinteknik, Eschenweg, Modautal, Germany). Periotestometry was repeated 3 times for each tooth and the mean values were shown in Table 1.

  1. Diagnostics.

4.1. Diagnosis of ankylosed teeth is a difficult task. The mentioned and used methods such as percussion, periotestometry and X-ray diagnostics are not reliable enough, since they are associated with the probability of establishing false positive or false negative conclusions. The most reliable method is considered to be the application of orthodontic force, but you have not provided information about this method. Please give a comment on this.

Thank you for your suggestion. We added the sentence for diagnosis of ankylosed teeth in the Diagnosis and Treatment procedures part.

The treatment objectives were: (1) to diagnose which tooth is actually ankylosed (2) to retract the ankylosed teeth via luxation, and (3) to achieve normal overjet and overbite with ideal occlusion. The treatment plan was as follows:

  1. Fixed rigid lingual arch devices were cemented on both the upper and lower jaws to assist with both anchorage and eruption of the ankylosed teeth.
  2. All teeth were moved orthodontically to diagnose the ankylosed teeth.

After initial leveling of the maxillary dentition with a 0.016-inch NiTi (SENTALLOY, TOMY INTERNATIONAL, Tokyo, Japan), the upper right central incisor, lower right lateral incisor, and lower right canine were diagnosed as ankylosed teeth, because they did not show tooth movement. Then, the tooth were luxated and a 0.014-inch NiTi (SENTALLOY, TOMY INTERNATIONAL, Tokyo, Japan) wire was used to retract the ankylosed teeth.

  1. Treatment.

The method used for replacing the missing tooth 33 is very interesting. Insufficient attention is paid to this aspect of treatment in the study. I could not find answers to the following questions in the text of the article:

5.1. Why such method of missing tooth replacement was used in the study? For example, the removable construction could be used as alternative method. Removable construction also would allow to keep the space between adjacent teeth for the further implantation.

Thank you for your suggestion. We added the explanation with a reference in the discussion.

To anchor the prosthesis, we applied direct bonding prosthesis, because patient’s satisfaction was greater than those with removable partial denture in young people. Because it is expected that vertical development of natural teeth in young female will occur even after adulthood, dental implant could not be applied at this time [27].

5.2. Are you going to replace the missing tooth with artificial one supported by dental implant in the future? Please complete the text with information about the long-term treatment plan.

Thank you very much for your advice. Actually, we planned to replace the prosthesis, so, we added the sentence as below.

However, we are going to replace this direct bonding prosthesis by dental implant after patient’s skeletal changes completely finish.

5.3. Why was the tooth 32 chosen as a supporting not tooth 34? From the point of view of masticatory force redistribution, tooth 34 or two adjacent teeth (34 and 32) are worth using as supporting structures. 5.4. In the text [lines 289-290] you indicated that tooth 34 was healthy so a tooth 32 was used as supporting. However, a tooth 32 was also healthy. Such logic is not clear enough for me. Please give additional arguments.

Thank you for your suggestion. We added the explanation with reference as below.

Pjetursson showed that survival rates of both dental implant and restorations in combined tooth-implant-supported prosthesis were lower than those in solely implant-supported prosthesis [28]. Hence, they recommended that prosthetic rehabilitation should be applied by solely implant-supported prosthesis. In this case, since the lower right lateral incisor was the ankylosed tooth, we decided to set direct bonding prosthesis as a single retainer because of the difference in degree of displacement between healthy right first premolar and ankylosed lateral incisor same as dental implants.

5.5. It is unclear how the reliability of the method of replacement of the missing tooth 33 is achieved. You have mentioned that the tooth is made of ceramic material that was simply attached to the lateral surface of the tooth 32. In my opinion, such adhesive bonding is not reliable. Have there been cases of artificial tooth debonding or fracturing? Which fixation protocol was used? Does tooth 33 participate in occlusive relationships? Please give comments on this.  5.6. In the text [line 291] you conclude that the artificial crown 33 allowed to restore the function, but the article does not provide any objective data confirming this fact. In my opinion, in order to reduce the risk of technical complications of such treatment, it would be necessary to remove the artificial tooth 33 from the occlusive relationship or significantly reduce the masticatory load that falls on it, which cannot be considered the restoration of masticatory function. The wording needs to be corrected.

Thank you for your suggestion. As you say, this direct bonding tooth did not participate in the occlusion. So, we added the sentence as below in the discussion.

The prosthetic device was esthetically pleasing and patient satisfaction was very high, although this device did not participate in the occlusive relationships in order to avoid debonding and fracturing. Therefore, we are going to replace this device by dental implant to improve occlusal function after patient’s skeletal changes completely finish.

5.7. Please comment on the post-treatment panoramic radiograph. How do you assess the short-term and long-term results of the treatment?

Thank you for your advice. We added the sentence in the discussion (last paragraph).

However, a long-term follow-up will be necessary for occlusal stability and root resorption of ankylosed teeth by radiograph examination.

  1. Conclusions.

Unfortunately, one clinical case does not provide grounds for formulating a conclusion about the effectiveness of treatment. The format of the clinical case is descriptive, in which the facts are simply stated. For example, in CARE (https://www.care-statement.org/checklist) there is no requirement to formulate conclusions, and therefore I would recommend deleting the section "Conclusions" from the article.

Thank you for your advice. We agree your suggestion. So, we deleted the conclusion section.

Reviewer 2 Report

Comments to the author:

Thank you for inviting me to review the paper entitled “Camouflage correction of skeletal Class III severe Open Bite with Tooth Ankyloses treated by temporary Anchorage devices: a case report”. The manuscript reports a case of a 14-year-old female patient with multiple teeth in ankylosis. The three teeth were luxated and pulled to the arch with orthodontic treatment (fixed appliance and TADs). The lower mandibular canine had to be extracted due to the impossibility to pull it to the arch. This is a very nice case report, nicely written, and explicative of how different the outcomes might be according to the possibility of moving or not an ankylosed tooth. It is very unique that this occurred in the same patient, and provided different modality of treatment. 

The major flaw is the lack of photographic documentation. There is only one photographic documentation taken at the bonding of the appliance. Unfortunately, this is too little of an information, considering that the case last 50 months and was not a straightforward case. Unless the authors can provide more pictures to document the phase, otherwise I would not leaning toward the acceptance of this case for publication. This is very unfortunate, as the case was very interesting and very well written. I hope the authors can provide more pictures to support the treatment evolution. 

A second major flaw is the lack of follow-up, unless the authors did not mention it in the paper.  

Minor corrections: 

- the definition of ankylosis in the introduction requires a citation. Same for the modalities to diagnosis and identify it. 

- I would look for data indicating which type of teeth are more predisposed to ankylosis. 

- I would remove the list of (1), (2), (3), … from lines 47 to 49, unless the authors are providing different references for each of the factors. 

- the ones indicated in lines 49-51 are “clinical signs”, not “clinical symptoms”. Please, correct the term

- Please, correct the different font of the text, to make it homogeneous throughout the manuscript 

- what does it mean in line 93 “she had bruised”? I am not sure what the authors referred to here, but I don’t believe it is a scientifically accepted dental expression

- the subsection of 2.4 Personal and Family History should not include the oral hygiene of the patient, as the oral hygiene is observed through clinical examination. Please, replace the statement of the oral hygiene (also detail it more, with information about bleeding on probing, deposit of plaque, periodontal pockets, etc) to the 2.5 Physical Exam.

- 2.5 should be named as Clinical Examination or Instrumental Examination, instead of Physical Examination. Why only those teeth were evaluated? Please, include in the table legend that only anterior teeth and premolars were examined. 

- panoramic radiograph: any comment on the relatively short roots of many of the teeth? Do the authors adduce this to the magnification of the x-ray or do they allude it to any associations with the ankyloses? The panoramic radiograph should also describe the TMJ, as best as possible despite being a 2-D image

 - which ceph analysis was utilized? Please, clarify that in the manuscript 

- line 145: lower right LATERAL incisor

- line 272: citation should be provided (after a number of successful cases have been reported). 

- why was the prosthetic device only cemented lingual to the lateral incisor, instead of connecting also to the premolar? Wouldn’t the authors incur in a rotation of the device? 

- any photographs or any note of follow-up visits? 

Author Response

April 7, 2023

Dear Editor, Dentistry Journal

I have now completed to revise and am now sending the manuscript entitled “Correction of skeletal Class III severe open bite with tooth ankyloses treated by temporary anchorage devices: A case report” (No: 2323350). I have revised the manuscript in accordance with the referee’s suggestions. A list of annotations is shown below. I hope the revised version will be well accepted by the editorial committee.

Thank you very much for all the troubles you've taken for me.

Sincerely Yours,

Yuka Yashima, Masato Kaku, Taeko Yamamoto, Cynthia Concepcion Medina, Shigehiro Ono, Yosuke Takeda and Kotaro Tanimoto

Address correspondence to: Masato Kaku, D.D.S., PhD.

Department of Anatomy and Functional Restorations,

Division of Oral Health Sciences,
Hiroshima University Graduate School of Biomedical and Health Sciences, Kasumi, Minami-ku, Hiroshima 734-8553, Japan.

E-mail: mkaku@hiroshima-u.ac.jp

To Reviewer 2

Thank you for inviting me to review the paper entitled “Camouflage correction of skeletal Class III severe Open Bite with Tooth Ankyloses treated by temporary Anchorage devices: a case report”. The manuscript reports a case of a 14-year-old female patient with multiple teeth in ankylosis. The three teeth were luxated and pulled to the arch with orthodontic treatment (fixed appliance and TADs). The lower mandibular canine had to be extracted due to the impossibility to pull it to the arch. This is a very nice case report, nicely written, and explicative of how different the outcomes might be according to the possibility of moving or not an ankylosed tooth. It is very unique that this occurred in the same patient, and provided different modality of treatment. 

The major flaw is the lack of photographic documentation. There is only one photographic documentation taken at the bonding of the appliance. Unfortunately, this is too little of an information, considering that the case last 50 months and was not a straightforward case. Unless the authors can provide more pictures to document the phase, otherwise I would not leaning toward the acceptance of this case for publication. This is very unfortunate, as the case was very interesting and very well written. I hope the authors can provide more pictures to support the treatment evolution. 

Thank you for your suggestion. We added the more pictures according to your advice (Figures 5A-C and 7A-C).

A second major flaw is the lack of follow-up, unless the authors did not mention it in the paper.  

Thank you for your suggestion. We added the sentence including follow-up treatment in the discussion part as below.

To anchor the prosthesis, we applied direct bonding prosthesis, because patient’s satisfaction was greater than those with removable partial denture in young people. Because it is expected that vertical development of natural teeth in young female will occur even after adulthood, dental implant could not be applied at this time [27]. Pjetursson showed that survival rates of both dental implant and restorations in combined tooth-implant-supported prosthesis were lower than those in solely implant-supported prosthesis [28]. Hence, they recommended that prosthetic rehabilitation should be applied by solely implant-supported prosthesis. In this case, since the lower right lateral incisor was the ankylosed tooth, we decided to set direct bonding prosthesis as a single retainer because of the difference in degree of displacement between healthy right first premolar and ankylosed lateral incisor same as dental implants. The prosthetic device was esthetically pleasing and patient satisfaction was very high, although this device did not participate in the occlusive relationships in order to avoid debonding and fracturing. Therefore, we are going to replace this device by dental implant to improve occlusal function after patient’s skeletal changes completely finish.

Minor corrections: 

- the definition of ankylosis in the introduction requires a citation. Same for the modalities to diagnosis and identify it. 

Thank you for your advice. We added the reference.

Hemley S. Fundamentals of occlusion. Munksgaard, Philadelphia, 1944, 184-192.

- I would look for data indicating which type of teeth are more predisposed to ankylosis. 

Thank you for your suggestion. We added the sentence with reference as below.

Dental ankylosis of permanent teeth is 10 times less frequently than primary teeth with mandibular and maxillary first molars being most frequently ankylosed followed by maxillary canines and incisors [2].

- I would remove the list of (1), (2), (3), … from lines 47 to 49, unless the authors are providing different references for each of the factors. 

Thank you for your advice. We removed the numbers from the text.

- the ones indicated in lines 49-51 are “clinical signs”, not “clinical symptoms”. Please, correct the term

Thank you for your suggestion. We corrected the term to clinical signs.

- Please, correct the different font of the text, to make it homogeneous throughout the manuscript 

Thank you for your suggestion. We corrected the font throughout the text.

- what does it mean in line 93 “she had bruised”? I am not sure what the authors referred to here, but I don’t believe it is a scientifically accepted dental expression

Thank you for your suggestion. We changed the sentence as below.

She fell on the ground and landed on her upper and lower right anterior teeth when she was 9 years old.

- the subsection of 2.4 Personal and Family History should not include the oral hygiene of the patient, as the oral hygiene is observed through clinical examination. Please, replace the statement of the oral hygiene (also detail it more, with information about bleeding on probing, deposit of plaque, periodontal pockets, etc) to the 2.5 Physical Exam.

Thank you for your advice. We replaced the statement of the oral hygiene to the physical examination with more information.

The patient’s oral hygiene was poor (plaque control record: 67%, probing pocket depth are less than 3 mm without bleeding on probing). So, continuous teeth brushing instruction was performed.

- 2.5 should be named as Clinical Examination or Instrumental Examination, instead of Physical Examination. Why only those teeth were evaluated? Please, include in the table legend that only anterior teeth and premolars were examined. 

Thank you for your advice. We changed the name 2.5 to Clinical examination.

Unfortunately, we applied periotestometry only on the teeth of open bite region. So, we added the sentence as below in the physical examination.

2.5. Clinical examination

To test the mobility of teeth around open bite region, a resonance frequency analyzer was used (Periotest, Gulden-Medizinteknik, Eschenweg, Modautal, Germany). Periotestometry was repeated 3 times for each tooth and the mean values were shown in Table 1.

- panoramic radiograph: any comment on the relatively short roots of many of the teeth? Do the authors adduce this to the magnification of the x-ray or do they allude it to any associations with the ankyloses? The panoramic radiograph should also describe the TMJ, as best as possible despite being a 2-D image

Thank you for your advice. We added the observation in Imaging examination as below.

2.6. Imaging examinations

A panoramic radiograph (Hyper-X, ASAHI ROENTGEN, Kyoto, Japan) revealed loss of the PDL cavity in the upper right central incisors, lower right lateral incisors, and lower right canine. Although the root of the upper and lower anterior tooth were relatively short, the association with the ankyloses is not clear. There was no abnormal condition both on the right and left temporomandibular joint (Figure 3).

 - which ceph analysis was utilized? Please, clarify that in the manuscript 

Thank you for your advice. We used the Hiroshima University cephalometric analyses, so, we added the sentence as below.

A skeletal Class III relationship of the angle of point A-nasion-point B (ANB; 0.2°) and a steep Frankfort-mandibular plane angle (FMA; 36.2°) with lingual inclination of the lower incisors of incisor-mandibular plane angle (IMPA; 79.5°) were noted in the Hiroshima University cephalometric analyses (CX-150W, ASAHI ROENTGEN, Kyoto, Japan, Figure 4 and Table 2).

- line 145: lower right LATERAL incisor

Thank you for your suggestion.

We changed lower right lateral incisors to lower right lateral incisor.

- line 272: citation should be provided (after a number of successful cases have been reported). 

Thank you for your advice. We added the more references (27-31).

- why was the prosthetic device only cemented lingual to the lateral incisor, instead of connecting also to the premolar? Wouldn’t the authors incur in a rotation of the device? 

Thank you for your suggestion. We added the explanation in the discussion as below.

To anchor the prosthesis, we applied direct bonding prosthesis, because patient’s satisfaction was greater than those with removable partial denture in young people. Because it is expected that vertical development of natural teeth in young female will occur even after adulthood, dental implant could not be applied at this time [27]. Pjetursson showed that survival rates of both dental implant and restorations in combined tooth-implant-supported prosthesis were lower than those in solely implant-supported prosthesis [28]. Hence, they recommended that prosthetic rehabilitation should be applied by solely implant-supported prosthesis. In this case, since the lower right lateral incisor was the ankylosed tooth, we decided to set direct bonding prosthesis as a single retainer because of the difference in degree of displacement between healthy right first premolar and ankylosed lateral incisor same as dental implants.

- any photographs or any note of follow-up visits? 

We are sorry that we do not have pictures of follow-up. We added the sentence in the discussion as below.

Therefore, it is concluded that the use of TADs was useful for retraction of ankylosed teeth and distalization of lower arch for camouflage skeletal Class III treatment without any adverse reactions. However, a long-term follow-up will be necessary for occlusal stability and root resorption of ankylosed teeth by radiograph examination.

Reviewer 3 Report

Introduction

"Ankylosis is a rare disorder characterized by the fusion of bony structures to one another."

I would not define ankylosis a rare disorder, please rephrase it.

"Dental ankyloses are defined as the obliteration of the periodontal ligament (PDL); thus, the involved tooth is fused to the surrounding bone, preventing eruption and orthodontic movement."

singular or plural?

The frequency of ankylosis occurrence is approximately 5 to 9.9%

In which population?

"Clinical symptoms include....."

The ones the authors mentioned seems clinical signs.

"In other words, it is challenging to detect punctate ankylosis using these methods"

The authors did not mention that this was the aim of the paraghraph. Moreover, it is not clear if the signs described are associated to every type of ankylosis.

"In this case report, we describe the results of treatment for open bite caused by tooth ankylosis using temporary anchorage devices (TADs), with no adverse reactions."

It is not clear the type of treatment. Is it the retraction after luxation? Please specifiy.

Methods

"The periotest values (PTV), which were measured using a resonance frequency analyzer (Periotest, Gulden-Medizinteknik, Eschenweg, Modautal, Germany), were found to be small in the upper right central incisor and the lower right canine."

It is not celar the scope and the mening of the Periotest values. Please give more detailed explanations.

"upper right central incisors, lower right lateral incisors"

It should be singular.

Table II: The authors mentioned the post-treatment values before explaining the treatment itself.

"Luxation of the ankylosed teeth was performed several times whenever the teeth showed ankyloses"

Describe the procedure of luxation

Discussion

The discussion is not exhaustive and not clear. The concepts expressed are reported in a confusionary manner.

Author Response

April 7, 2023

Dear Editor, Dentistry Journal

I have now completed to revise and am now sending the manuscript entitled “Correction of skeletal Class III severe open bite with tooth ankyloses treated by temporary anchorage devices: A case report” (No: 2323350). I have revised the manuscript in accordance with the referee’s suggestions. A list of annotations is shown below. I hope the revised version will be well accepted by the editorial committee.

Thank you very much for all the troubles you've taken for me.

Sincerely Yours,

Yuka Yashima, Masato Kaku, Taeko Yamamoto, Cynthia Concepcion Medina, Shigehiro Ono, Yosuke Takeda and Kotaro Tanimoto

Address correspondence to: Masato Kaku, D.D.S., PhD.

Department of Anatomy and Functional Restorations,

Division of Oral Health Sciences,
Hiroshima University Graduate School of Biomedical and Health Sciences, Kasumi, Minami-ku, Hiroshima 734-8553, Japan.

E-mail: mkaku@hiroshima-u.ac.jp

To Reviewer 3

Introduction

"Ankylosis is a rare disorder characterized by the fusion of bony structures to one another."

I would not define ankylosis a rare disorder, please rephrase it.

Thank you for your suggestion.

We changed the sentence as below.

Tooth ankylosis is generally characterized by the fusion of bony structures to one another.

"Dental ankyloses are defined as the obliteration of the periodontal ligament (PDL); thus, the involved tooth is fused to the surrounding bone, preventing eruption and orthodontic movement."

singular or plural?

Thank you for your advice. We corrected the sentence as below.

Tooth ankylosis is generally characterized by the fusion of bony structures to one another. This disorder is defined as the obliteration of the periodontal ligament (PDL); thus, the involved tooth is fused to the surrounding bone, preventing eruption and orthodontic movement.

The frequency of ankylosis occurrence is approximately 5 to 9.9%

In which population?

Thank you for your suggestion. We changed the sentence and reference as below.

The frequency of ankylosis occurrence is 22.4% in Sweden.

"Clinical symptoms include....."

The ones the authors mentioned seems clinical signs.

Thank you for your advice. We changed “clinical symptoms” to “clinical signs”.

"In other words, it is challenging to detect punctate ankylosis using these methods"

The authors did not mention that this was the aim of the paragraph. Moreover, it is not clear if the signs described are associated to every type of ankylosis.

Thank you for your suggestion. We omitted this sentence.

"In this case report, we describe the results of treatment for open bite caused by tooth ankylosis using temporary anchorage devices (TADs), with no adverse reactions."

It is not clear the type of treatment. Is it the retraction after luxation? Please specifiy.

Thank you for your advice. We added the information as below.

In this case report, we retracted ankylosed tooth using temporary anchorage devices (TADs) after luxation, with no adverse reactions.

Methods

"The periotest values (PTV), which were measured using a resonance frequency analyzer (Periotest, Gulden-Medizinteknik, Eschenweg, Modautal, Germany), were found to be small in the upper right central incisor and the lower right canine."

It is not celar the scope and the mening of the Periotest values. Please give more detailed explanations.

Thank you for your suggestion. We added the explanations as below.

To test the mobility of teeth around open bite region, a resonance frequency analyzer was used (Periotest, Gulden-Medizinteknik, Eschenweg, Modautal, Germany). Periotestometry was repeated 3 times for each tooth and the mean values were shown in Table 1. The periotest values (PTV) were found to be small in the upper right central incisor and the lower right canine and lateral incisor.

"upper right central incisors, lower right lateral incisors"

It should be singular.

Thank you for your suggestion. We corrected to “upper right central incisor, lower right lateral incisor”.

Table II: The authors mentioned the post-treatment values before explaining the treatment itself.

We are sorry we do not understand what we should do.

"Luxation of the ankylosed teeth was performed several times whenever the teeth showed ankyloses"

Describe the procedure of luxation

Thank you for your suggestion. We described the procedure of luxation as below.

Surgical luxation of the ankylosed teeth was performed using anterior after local anesthesia. Luxation of the teeth was repeated several times whenever the teeth showed ankylosis.

Discussion

The discussion is not exhaustive and not clear. The concepts expressed are reported in a confusionary manner.

Thank you for your advice. We added more discussion with references.

To anchor the prosthesis, we applied direct bonding prosthesis, because patient’s satisfaction was greater than those with removable partial denture in young people. Because it is expected that vertical development of natural teeth in young female will occur even after adulthood, dental implant could not be applied at this time [27]. Pjetursson showed that survival rates of both dental implant and restorations in combined tooth-implant-supported prosthesis were lower than those in solely implant-supported prosthesis [28]. Hence, they recommended that prosthetic rehabilitation should be applied by solely implant-supported prosthesis. In this case, since the lower right lateral incisor was the ankylosed tooth, we decided to set direct bonding prosthesis as a single retainer because of the difference in degree of displacement between healthy right first premolar and ankylosed lateral incisor same as dental implants. The prosthetic device was esthetically pleasing and patient satisfaction was very high, although this device did not participate in the occlusive relationships in order to avoid debonding and fracturing. Therefore, we are going to replace this device by dental implant to improve occlusal function after patient’s skeletal changes completely finish.

Round 2

Reviewer 1 Report

Dear authors!

The manuscript has been modified according to the suggestions and improved according to my point of view.

Reviewer 2 Report

Thank you to the authors for providing a updated version of their manuscript. In this new version, the authors diligently addressed all the previous comments and concerns of the reviewers. My major concern is the lack of photograph documentation of the case (during the treatment to document the most significant phase and in the retention phase). However, if the associate editor considered the current documentation enough for publication, I stay at his/her ultimate decision.  

Minor points:
- line 64, I suggest that the authors clearly state the aim of the case report (something like, "in this case report, we aim at describing a case of ankylosed tooth resolved by using ...").  
- as asked in my previous comment, the author needs to clarify when the post-treatment pictures were taken. Was it right at the debonding procedure?  

Author Response

April 12, 2023

Dear Editor, Dentistry Journal

I have now completed to revise and am now sending the manuscript entitled “Correction of skeletal Class III severe open bite with tooth ankyloses treated by temporary anchorage devices: A case report” (No: 2323350). I have revised the manuscript in accordance with the referee’s suggestions. A list of annotations is shown below. I hope the revised version will be well accepted by the editorial committee.

Thank you very much for all the troubles you've taken for me.

Sincerely Yours,

Yuka Yashima, Masato Kaku, Taeko Yamamoto, Cynthia Concepcion Medina, Shigehiro Ono, Yosuke Takeda and Kotaro Tanimoto

Address correspondence to: Masato Kaku, D.D.S., PhD.

Department of Anatomy and Functional Restorations,

Division of Oral Health Sciences,
Hiroshima University Graduate School of Biomedical and Health Sciences, Kasumi, Minami-ku, Hiroshima 734-8553, Japan.

E-mail: mkaku@hiroshima-u.ac.jp

To Reviewer 2

Minor points:
- line 64, I suggest that the authors clearly state the aim of the case report (something like, "in this case report, we aim at describing a case of ankylosed tooth resolved by using ...").  

Thank you for your advice. We changed the sentence as below.

In this case report, we aim at describing a case of ankylosed tooth resolved by using temporary anchorage devices (TADs) after luxation, with no adverse reactions.

- as asked in my previous comment, the author needs to clarify when the post-treatment pictures were taken. Was it right at the debonding procedure?  

Thank you for your suggestion. We added the sentence in the Treatment procedures, Results, and legend of figure 9 as below.

Treatment procedures

Surgical luxation of the ankylosed teeth was performed using anterior after local anesthesia. Luxation of the teeth was repeated several times whenever the teeth showed ankylosis. However, because the lower right canine erupted only half-way to the occlusal plane, the tooth was extracted and after healing the periodontal tissue, the resulting space was closed with a ceramic crown using the right lateral incisor as a single retainer.

Results

After orthodontic treatment, the arch alignment was well corrected, and the overbite increased to 2.2 mm, and the overjet improved to 2.0 mm. There was no major change in the post-treatment facial profile compared to the pretreatment profile. The left molar relationship was Angle class I, whereas that on the right was Angle class III. The space of lower right canine was repaired by direct bonding right after debonding of all orthodontic applliances.

legend of figure 9

Post-treatment intraoral photographs right after debonding all orthodontic appliances.

Reviewer 3 Report

Thank you for the corrections to the manuscript

Author Response

April 12, 2023

Dear Editor, Dentistry Journal, and Review 3

Thank you for your important suggestion. I have now completed to revise and am now sending the manuscript entitled “Correction of skeletal Class III severe open bite with tooth ankyloses treated by temporary anchorage devices: A case report” (No: 2323350). I have revised the manuscript in accordance with the referee’s suggestions. I hope the revised version will be well accepted by the editorial committee.

Thank you very much for all the troubles you've taken for me.

Sincerely Yours,

Yuka Yashima, Masato Kaku, Taeko Yamamoto, Cynthia Concepcion Medina, Shigehiro Ono, Yosuke Takeda and Kotaro Tanimoto

Address correspondence to: Masato Kaku, D.D.S., PhD.

Department of Anatomy and Functional Restorations,

Division of Oral Health Sciences,
Hiroshima University Graduate School of Biomedical and Health Sciences, Kasumi, Minami-ku, Hiroshima 734-8553, Japan.

E-mail: mkaku@hiroshima-u.ac.jp